# Clinical Characteristics of 6102 Asymptomatic and Mild Cases for Patients with COVID-19 in Indonesia

Erwin Astha Triyono [1,*], Joni Wahyuhadi [2], Christijogo Soemartono Waloejo [3], Dimas Aji Perdana [1], Nabilah [1], Sisilia Dewanti [1], Amal Arifi Hidayat [1], Michael Austin Pradipta Lusida [1], Fani Sarasati [4], Ngurah Arie Kapindra Dharma [4], Muhammad Ikhtiar Zaki Al Razzak [4], Tanri Hadinata Wiranegara [4] and Nurarifah Destianizar Ali [4]

[1] Department of Internal Medicine, Faculty of Medicine, Dr. Soetomo Teaching Hospital, Universitas Airlangga, Surabaya 60286, Indonesia; dr_komando@yahoo.co.id (D.A.P.); nabilah.unair@yahoo.com (N.); sisil.dewanti@gmail.com (S.D.); arifiamal@gmail.com (A.A.H.); michaellusida@gmail.com (M.A.P.L.)

[2] Department of Neurosurgery, Faculty of Medicine, Dr. Soetomo Teaching Hospital, Universitas Airlangga, Surabaya 60286, Indonesia; joni.wahyuhadi@yahoo.com

[3] Department of Anesthesiology, Faculty of Medicine, Dr. Soetomo Teaching Hospital, Universitas Airlangga, Surabaya 60286, Indonesia

[4] Indrapura Forefront Hospital Surabaya, Surabaya 60175, Indonesia; ariekapindrangurah@yahoo.co.id (N.A.K.D.); muhammad.ikhtiar.zaki94@gmail.com (M.I.Z.A.R.); tan26691@gmail.com (T.H.W.); nurarifahda@gmail.com (N.D.A.)

\* Correspondence: erwintriyono@yahoo.com

**Abstract:** Background: The COVID-19 pandemic has led to a rise in confirmed cases, making epidemiological studies crucial for identifying the source of transmission and developing effective treatment methods. We conducted a study on the clinical characteristics of patients with asymptomatic and mild symptoms of COVID-19 at a rescue hospital in Indonesia. Methods: This is an epidemiological study involving 6102 patients who were admitted to the Indrapura forefront hospital in Surabaya from May 2020 to February 2021. We described demographic data, clinical signs and symptoms, laboratory data, therapy, and clinical outcomes. Results: A total of 6102 patients were involved in this study, with 3664 (60.04%) being male and 2438 (39.95%) being female. The age range of 21–30 years was the most prevalent, accounting for 31.1% (1898 patients). The population had 1476 patients (24.2%) with comorbid conditions. The most prevalent comorbidity observed among these patients was hypertension, affecting 1015 individuals (16.6%). Out of the total 6006 patients observed, 40.7% ($n = 2486$) were asymptomatic, 54.6% ($n = 3329$) had mild symptoms, and 3.1% ($n = 191$) had moderate symptoms. All patients were administered supportive therapy without the use of antiviral medication. Out of the 6102 patients included in the study, 5923 patients (97.1%) achieved a cure, 36 patients (0.6%) are currently undergoing treatment, 142 patients (2.3%) were referred for desaturation indications (SpO2 < 94%), and one patient died due to a suspected cardiovascular event. Out of the total number of patients, 74.5% (4529 patients) had an average length of stay (LOS) of less than 10 days, while 25.6% (1563 patients) had an average length of stay of more than 10 days. Conclusion: The clinical presentation of asymptomatic and mild COVID-19 patients at a rescue hospital varies significantly based on the age and sex of patients. Cough and hyposmia are commonly observed symptoms. Supportive therapy is effective, and strict implementation of social distancing is crucial in preventing the spread of this disease from individuals who are asymptomatic or have mild symptoms.

**Keywords:** COVID-19; asymptomatic; epidemiologic; clinical manifestation

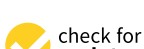



## 1. Introduction

Coronavirus disease (COVID-19), a global public health concern, was declared a pandemic in March 2020. COVID-19 is caused by a novel coronavirus known as 2019-nCoV, as defined by the World Health Organization (WHO) [1]. The novel coronavirus has been

formally identified as "Severe Acute Respiratory Syndrome Coronavirus 2" ('SARS-CoV-2') by the International Committee on Taxonomy of Viruses [2]. The patient presents clinical symptoms such as fever, cough, breathing difficulties, muscle pain, fatigue, normal or decreased leukocyte count, and radiographic findings consistent with pneumonia. Severe cases can lead to organ dysfunction, including shock, acute respiratory distress syndrome (ARDS), acute heart injury, acute kidney injury, and even death [3]. As of September 2020, there were 30,675,675 confirmed cases of COVID-19 reported in over 216 countries, resulting in 954,417 deaths and a mortality rate of 3.1%. Indonesia has reported a cumulative total of 248,852 COVID-19 cases, with a mortality rate of 9.677%. In the province of East Java, there have been 40,708 cases, accounting for 16.35% of the national total [4].

The ongoing increase in COVID-19 cases in certain countries necessitates the importance of conducting epidemiological studies to identify the source of transmission. The current understanding of COVID-19 epidemiology suggests that the primary mode of transmission for SARS-CoV-2 is through droplets and close contact with infected individuals [5]. Epidemiological studies can inform government strategies for managing and controlling disease transmission by identifying high-risk populations and recommending their localization in areas with a high prevalence of cases. As research on COVID-19 progresses, there remain uncertainties regarding patient epidemiology, including hospitalization duration and recovery outcomes. These factors are crucial for assessing the preparedness of the healthcare system [6].

Despite the World Health Organization declaring COVID-19 as a global public health emergency, there continues to be a rising global incidence of COVID-19 infections among patients. According to the existing literature, approximately 80% of individuals infected with COVID-19 experience mild symptoms [7]. The majority of studies have primarily examined patients with severe clinical symptoms, while only a limited number have specifically investigated the clinical progression of individuals with asymptomatic or mild cases of COVID-19 infection. There is limited systematic analysis available on asymptomatic SARS-CoV-2 infections. This study considers reported infections, prediction models, and undocumented or asymptomatic infections to inform strategies aimed at reducing transmissions [8]. This study aims to provide a description of the clinical characteristics of patients at a COVID-19 center in Indonesia.

## 2. Methodology

### 2.1. Study Design

This is a descriptive study of COVID-19 patients admitted to the Kogabwilhan II Indrapura forefront hospital in Surabaya, East Java, Indonesia, between May 2020 and February 2021. The data were acquired from electronic medical records using the Inova Medika Solusindo application. It included demographic information, clinical signs and symptoms, laboratory results, treatment details, and clinical outcomes.

### 2.2. Patient Criteria

All COVID-19 patients included in this study were diagnosed based on the diagnostic criteria outlined in the interim guidance for the clinical management of COVID-19 provided by the World Health Organization on 27 May 2020. All patients tested positive for SARS-CoV-2 infection through laboratory confirmation using specific RT-PCR SARS-CoV-2 tests. Mild-case patients were diagnosed by meeting the COVID-19 case definition and symptomatic criteria without evidence of viral pneumonia or hypoxia. Moderate cases were diagnosed in patients with clinical symptoms of pneumonia (fever, cough, dyspnea, and rapid breathing) but without signs of severe pneumonia, specifically SpO2 $\geq$ 90% in room air. The sample for this study consists of all patients treated at the Indrapura forefront hospital who meet the specified criteria.

*2.3. Data Characteristics*

Patient demographic data were collected from electronic medical records using a form. The collected data include age, gender, occupation, and place of residence. The collected clinical symptom data include information on signs and symptoms such as fever, cough, shortness of breath, fatigue, anorexia, muscle aches, headaches, chills, nausea and vomiting, diarrhea, and confusion. It also includes medical history details, including hypertension, heart disease, diabetes, obesity, chronic obstructive pulmonary disease, and liver disease. Additionally, physical examination data on vital signs such as blood pressure, pulse, respiratory rate, body temperature, oxygen saturation, and body mass index were documented. The disease's date of onset is determined as the initial day when symptoms became apparent.

Naso/oropharynx swabs were collected on the day of arrival, and RT-PCR analysis was conducted on SARS-CoV-2 data. The evaluation of the data was performed after the patient's discharge from the hospital. The collected clinical outcome data encompass recovery rates, treatment duration, referrals, mortality cases, and instances of self-quarantine.

*2.4. Clinical Management*

Symptomatic treatment was done by administering appropriate therapy, such as antipyretics, anti-diarrhea, decongestant, and antitussive therapies, according to the patient's complaint

Isolation was brought about by separating patients into red zones and separating patients from service providers. The room contained a chamber, a resting area, a field for outdoor activities, and a garden.

Relaxation was achieved by engaging in gymnastics activities outdoors and practicing deep breathing exercises. Furthermore, individuals could engage in musical activities while adhering to mask wearing and maintaining a safe distance. This program offered regular lectures on spirituality and religion, as well as stress management techniques.

The dietary intake was tailored to meet the specific caloric requirements of the patient, taking into consideration any existing medical conditions. The food provided was designed to meet the body's macro nutritional needs, including carbohydrates, proteins, fats, vitamins, and minerals.

Observation was conducted in three shifts, with each shift spending three hours in the isolation zone to monitor patients' vital signs, assess their ongoing complaints, and track their progress. During the observation period, patients had the opportunity to discuss various physical and mental issues, including sleep disturbances and feelings of restlessness. Patients had the option to communicate with the care team through online channels at any given time.

Overall, patients did not receive antibiotics or antiviral therapy.

*2.5. Statistical Analysis*

The statistical analysis was performed using SPSS Version 24. Continuous data are reported as mean $\pm$ standard deviation (SD). Categorical data are presented using numerical values and percentages.

**3. Result**

*3.1. Clinical Characteristics*

During the research period of May 2020 to February 2021, a cohort of 6102 patients was identified based on the predefined research criteria. Out of the total number of patients, 3664 (60%) were male. The most prevalent age range among the study subjects was 21–30 years, accounting for 31.1% (1898 patients). The age ranges of 31–40 years, 41–50 years, 11–20 years, 0–10 years, and above 61 years accounted for 23.35%, 20.40%, 9.59%, 1.02%, and 2.46%, respectively. The mean age of asymptomatic patients was 34.6 $\pm$ 12.5 years, while the mean age of patients with mild symptoms was 35.9 $\pm$ 12.9 years. A total of 4626 patients (75.8%) received treatment without comorbidities, while 1476 pa-

tients (24.2%) had comorbidities. The prevalent comorbidities included hypertension in 1015 patients (16.6%), diabetes mellitus in 272 patients (4.5%), obesity grade 1 in 1958 patients (32.1%), and obesity grade 2 in 997 patients (16.3%). A minority of patients exhibit comorbidities, including chronic kidney disease, bronchial asthma, and HIV. The study included a total of 3007 patients, with the majority (49.3%) being private employees. Civil servants accounted for 839 patients (13.7%), while military and police personnel made up 396 patients (6.5%). Students represented 387 patients (6.3%), teachers 139 patients (2.3%), and medical personnel 251 patients (4.1%). Among the medical personnel, there were 140 nurses (2.3%), 75 doctors (1.2%), and 36 midwives (0.6%) (Table 1).

**Table 1.** Demographic characteristics of patients at Indrapura Region II Joint Command Field Hospital.

| Patients Demographics (*n* = 6102) | Value (%) |
|:---:|:---:|
| Gender | |
| Male | 3664 (60%) |
| Female | 2438 (40%) |
| Age | |
| 0–10 years | 62 (1.02%) |
| 11–20 years | 585 (9.59%) |
| 21–30 years | 1898 (31.1%) |
| 31–40 years | 1434 (23.5%) |
| 41–50 years | 1245 (20.40%) |
| 51–60 years | 728 (11.93%) |
| >61 years | 150 (2.46%) |
| Occupation | |
| Private Employee | 3007 (49.3%) |
| Civil Servant | 839 (13.7%) |
| Military and Police | 396 (6.5%) |
| Student | 387 (6.3%) |
| Teacher | 139 (2.3%) |
| Healthcare Worker | |
| Nurse | 140 (2.3%) |
| Doctor | 75 (1.2%) |
| Midwife | 36 (0.6%) |
| Comorbidity | |
| No Comorbid | 4626 (75.8%) |
| Hypertension | 1015 (16.6%) |
| Diabetes Mellitus | 272 (4.5%) |
| Grade 1 Obesity | 1958 (32.1%) |
| Grade 2 Obesity | 997 (16.3%) |
| Others (CKD, asthma, HIV) | |

CKD: chronic kidney disease; HIV: human immunodeficiency virus.

### 3.2. Signs and Symptoms

Out of the COVID-19 patients treated at Indrapura forefront hospital, 40.7% (*n* = 2486) were asymptomatic, 54.6% (*n* = 3329) had mild symptoms, 3.1% (*n* = 191) had moderate symptoms, and 1.6% (*n* = 96) had severe symptoms. The predominant symptoms observed in COVID-19 patients were coughing (29.8%), fever (17.1%), anosmia (14.9%), common cold (14.6%), headache (8.1%), nausea (5.3%), shortness of breath (4.5%), diarrhea (2.4%), and muscle pain (2.4%). In addition, patients reported other symptoms, including abdominal pain in 118 cases (1.95%) and sleep disorders in 38 cases (0.6%) (Table 2).

**Table 2.** Clinical characteristics of patients at Indrapura Region II Joint Command Field Hospital.

| Characteristics (*n* = 6102) | Value (%) |
|---|---|
| Severity | |
| No Symptoms | 2486 (40.7%) |
| Mild | 3329 (54.6%) |
| Moderate | 191 (3.1%) |
| General Symptoms | |
| Cough | 1818 (29.8%) |
| Fever | 1046 (17.1%) |
| Anosmia | 911 (14.9%) |
| Cold | 889 (14.6%) |
| Headache | 493 (8.1%) |
| Nausea | 323 (5.3%) |
| Dyspnea | 273 (4.5%) |
| Diarrhea | 149 (2.4%) |
| Myalgia | 149 (2.4%) |
| Abdominal Pain | 118 (1.95%) |
| Sleep Disturbance | 38 (0.6%) |
| Swab RT-PCR SARS-CoV-2 | |
| Positive | 6102 (100%) |
| Feedback post-treatment | |
| Did not perform the re-swab | 1097 (18%) |
| Performed the re-swab | 5003 (82%) |
| Positive | 3124 (51.2%) |
| Negative | 1879 (30.8%) |

RT-PCR SARS-CoV-2: Reverse-Transcriptase Polymerase Chain Reaction Severe Acute Respiratory Syndrome Coronavirus 2.

*3.3. Swab RT-PCR SARS-CoV-2 Results*

All patients were admitted to the Indrapura forefront hospital after testing positive for SARS-CoV-2 using an RT-PCR swab. Recovered patients are individuals without the need for symptomatic medication who experienced a reduction or absence of clinical symptoms following treatment. Additionally, these patients tested negative for the virus on two separate occasions. Following the patient's discharge from the hospital, they offer feedback regarding the post-treatment swab examination. Out of the total number of patients, 82% (*n* = 5003) underwent a re-examination swab after being discharged from the Indrapura forefront hospital, while 18% (*n* = 1097) did not undergo this procedure. Out of the 5003 patients who underwent the swab test, 3124 patients (51.2%) tested positive, while 1879 patients (30.8%) tested negative (Table 2).

*3.4. Therapy*

Patients at the Indrapura forefront hospital received symptomatic therapy, including antitussive, expectorant, antipyretic, decongestant, and bronchodilator medications. They were also provided with isolation, relaxation, nutrition, and observation. Furthermore, patients received therapy to address comorbid conditions. In February 2021, the majority of patients (95.8%) received vitamin C therapy, while a smaller proportion (0.555%) received multivitamin therapy. Additionally, a significant number of patients (31.5%) were treated with N-acetyl cysteine 200 mg (NAC), followed by decongestant therapy for 15.4% of patients, paracetamol for 18.1% of patients, and lorazepam for anxiety disorders in 4.7% of patients. Among the comorbid hypertensive patients, 870 individuals (14.3%) were prescribed Amlodipine 10 mg, while 129 patients (2.1%) were administered Candesartan. Among the patients with comorbid diabetes, 82 (1.3%) received metformin 500 mg, 106 (1.7%) were prescribed Glimepiride 2 mg, and 88 (1.4%) were treated with insulin (specifically Apidra, Novorapid, and Levemir). A total of 158 patients (2.6%) received oxygen therapy due to desaturation. No patients at the Indrapura forefront hospital were administered antiviral therapy, antibiotics, or corticosteroids (Table 3).

**Table 3.** Characteristics of therapy and clinical outcomes of patients at Indrapura Region II Joint Command Field Hospital.

| Characteristics (*n* = 6102) | Value (%) |
|---|---|
| Symptomatic Therapy | |
| Vitamin C | 5846 (95.8%) |
| Multivitamins | 226 (0.555%) |
| N-acetylcystein 200 mg | 1921 (31.5%) |
| Decongestants (Pseudoephedrine 60 mg) | 940 (15.4%) |
| Paracetamol | 1107 (18.1%) |
| Lorazepam | 289 (4.7%) |
| Comorbid Therapy | |
| Amlodipine 10 mg | 870 (14.3%) |
| Candesartan | 129 (2.1%) |
| Metformin 500 mg | 82 (1.3%) |
| Glimepiride 2 mg | 106 (1.7%) |
| Insulin (Apidra, Novorapid, Levemir) | 88 (1.4%) |
| Oxygen Therapy | 158 (2.6%) |
| Clinical Outcomes | |
| Cured | 5923 (97.1%) |
| Self Quarantine | 36 (0.6%) |
| Referred | 142 (2.3%) |
| Died | 1 (0%) |
| Length of Treatment | |
| <10 days | 4529 (74.5%) |
| 10 days | 692 (11.3%) |
| >10 days | 871 (14.3%) |

*3.5. Clinical Outcomes*

Among the 6102 patients included in the study, the majority, specifically 5923 patients (97.1%), successfully recovered. A smaller proportion, specifically 36 patients (0.6%), chose to continue self-quarantine. A total of 142 patients, accounting for 2.3% of the sample, were referred due to indications of desaturation, defined as a peripheral oxygen saturation (SpO2) level below 94%. One patient (0%) died due to a cardiovascular event. The majority of patients, 74.5%, had a length of stay (LOS) of less than 10 days, with a total of 4529 patients. There were 871 patients, or 14.3%, who had an LOS of more than 10 days. Additionally, 692 patients, or 11.3%, had an LOS of 10 days (Table 3). The asymptomatic group and mild symptom group had average lengths of stay of 2.7 ± 0.7 and 2.5 ± 0.7 days, respectively. Those presenting with moderate symptoms were 17.3% asymptomatic and 82.7% mildly ill at the beginning of hospitalization. Additionally, it was observed that 46 out of 191 patients (24.1%) who initially presented with moderate symptoms progressed to severe disease following admission.

**4. Discussion**

The World Health Organization (WHO) has classified SARS-CoV-2 infection, commonly referred to as COVID-19, as a global public health issue or pandemic since March 2020 [9]. Clinical manifestations of COVID-19 patients varied, with supporting and clinical outcomes also very different. In this study, a total of 6102 patients were reported to be treated at the Indrapura forefront hospital from May 2020 to February 2021. A total of 3664 patients (60%) were found to be male, and the age of most of the patients treated, 1898 patients (31.1%), was between 21 and 30 years. A total of 3007 patients (49.3%) worked as private employees, and 251 patients (4.1%) as medical professionals. These data are similar to research in the Indrapura forefront hospital in 2020, which found that the patients treated are predominantly male and of productive age. Retrospective research on 256 COVID-19 patients in Zhejiang in 2020 stated that of the 234 symptomatic patients, 62.4% were men with an average age of 48 years. The average age of asymptomatic pa-

tients was significantly lower than patients with symptoms (33.0 years versus 47.6 years, *p* < 0.001) [10].

Obesity was the most prevalent comorbidity, observed in 48.4% of the total patient population (*n* = 2955). Hypertension was the second most common comorbidity, affecting 16.6% of patients (*n* = 1015), while diabetes mellitus was present in 4.5% of patients (*n* = 272). The data from the research conducted at Indrapura Forefront Hospital between 28 May and 20 September 2020 reveal that hypertension was the most common comorbidity, affecting 286 patients (13.5%), followed by diabetes mellitus, which affected 84 patients (4%). Research conducted in Saudi Arabia showed a significant association between COVID-19 and obesity, with 64% of patients having comorbid obesity [11]. A study conducted in China investigated the clinical characteristics of 41 patients infected with COVID-19. The findings revealed that 13 cases (32%) had underlying health conditions, including hypertension, chronic obstructive pulmonary disease, diabetes, and cardiovascular disease [12]. A systematic review identified a correlation between obesity and unfavorable outcomes in individuals diagnosed with COVID-19. The assessment of comorbidities and BMI measurements is recommended in the treatment of COVID-19 patients [13].

At the Indrapura forefront hospital, a total of 2486 patients (40.7%) were asymptomatic, 3329 patients (54.6%) had mild symptoms, 191 patients (3.1%) had moderate symptoms, and 96 patients (1.6%) had severe symptoms. The predominant symptoms observed in COVID-19 patients include cough (29.8%), fever (17.1%), anosmia (14.9%), and common cold symptoms (14.6%) in a sample of 1818, 1046, 911, and 889 patients, respectively. The data are similar to the preliminary research conducted in 2020, which identified 1238 patients without complaints (58.3%). The remaining patients experienced mild symptoms, including cough (15.3%), common cold (6.2%), anosmia (5.1%), and fever (4.3%). The World Health Organization (WHO) in 2021 reported several common symptoms of COVID-19. These include fever (83–99%), cough (59–82%), fatigue (44–70%), anorexia (40–84%), shortness of breath (31–40%), and myalgia (11–35%) [14]. Pre-onset respiratory symptoms have been associated with the occurrence of anosmia (loss of smell) or ageusia (loss of taste). Elderly and immunosuppressed individuals may exhibit non-typical symptoms, including fatigue, diminished alertness, limited mobility, diarrhea, decreased appetite, and confusion [15]. According to Chen et al., mild symptomatic pneumonia is the predominant cause [16]. The reported findings differ from those observed in severe symptoms, where fever commonly occurs. Therefore, it is important to monitor patients with fever complaints for potential clinical deterioration [17].

Besides testing symptomatic individuals, the COVID-19 tests were also administered to asymptomatic individuals who have had close contact with confirmed cases in order to mitigate transmission rates. The treatment administered to asymptomatic COVID-19 patients is supportive and symptomatic [18]. In addition to symptomatic therapy, patients at Indrapura Joint Command Field Hospital II received isolation, relaxation, nutrition, observation, and comorbid therapy. Antipyretics for fever and pain and antitussive, expectorant, decongestant, bronchodilator, and rehydration treatments are all forms of symptomatic treatment. Patients with a history of comorbidities are advised to continue taking their previous medications based on their clinical condition.

Counseling is necessary for patients with mild symptoms to address potential complications that require prompt reporting for further treatment in addition to symptomatic therapy [19]. Patients suspected or confirmed to have mild COVID-19 should be promptly isolated in accordance with the established COVID-19 treatment protocol, as recommended by the World Health Organization (WHO), in order to prevent the transmission of the virus. COVID-19 patients can receive care in designated health facilities, community facilities, or at home through self-isolation with observation [14].

Vitamin C and vitamin D are potential multivitamins for COVID-19 patients. Hoang et al.'s research demonstrates the potential advantages of vitamin C supplementation as an antioxidant in the context of viral infections [20]. A meta-analysis was conducted to examine the impact of vitamin D supplementation on acute respiratory tract infections.

The findings indicate that regular supplementation of vitamin D2/D3, at a dosage of up to 2000 IU/day, provides protection against such infections [21]. Therefore, vitamin C and D supplementation may be administered to individuals diagnosed with COVID-19.

Several antiviral drugs were available for the treatment of patients with COVID-19. The efficacy and safety of antiviral drugs such as Chloroquine, Hydroxychloroquine, Oseltamivir, Lopinavir/Ritonavir, Favipiravir, and Remdesivir in managing COVID-19 have been investigated in small-scale clinical studies. However, these studies are insufficient to establish definitive conclusions regarding their effectiveness and safety [22]. Pruijssers et al. conducted a study which found no clinical benefits associated with Remdesivir treatment. The study was terminated prematurely due to an observed rise in side effects [23]. The World Health Organization (WHO) does not endorse the routine utilization of Remdesivir for hospitalized patients with COVID-19. The World Health Organization advises against the use of Hydroxychloroquine, Chloroquine, and Lopinavir/Ritonavir for the treatment of COVID-19 [24].

Prophylactic administration of antibiotics to COVID-19 patients with mild symptoms is not advised due to the potential for increased bacterial resistance, which can subsequently exacerbate the morbidity and mortality associated with the disease [25]. Severe COVID-19 patients who are at risk of bacterial infection may receive empirical antibiotic treatment in accordance with the guidelines provided by the World Health Organization (WHO). Antibiotics should be administered within one hour of the initial assessment, preferably after obtaining a culture sample [14].

The WHO's fourth living guideline reviewed the utilization of ivermectin in individuals diagnosed with COVID-19. This coincided with the growing global interest in ivermectin as a potential therapeutic intervention. Although there is ongoing research on the use of ivermectin for prophylaxis, the guideline specifically focuses on its therapeutic role in the treatment of COVID-19. The evidence summary included data from 16 trials and 2407 participants. The network meta-analysis provided relative estimates of effect for patient-important outcomes. Among the trials analyzed, 75% focused on patients with non-severe disease, while 25% encompassed both severe and non-severe patients. Several trials did not provide data on the outcomes we were interested in [24].

Systemic corticosteroids are administered to COVID-19 patients with severe symptoms in accordance with the recommendations provided by the World Health Organization (WHO) [26]. Additional therapeutic modalities, including Immunoglobulin (Ig), convalescent plasma (CP), IL-6 inhibitors, and TNF alpha inhibitors, may be beneficial in specific patient populations depending on the severity of the disease. It is imperative to individualize treatment options and closely monitor outcomes, particularly due to the limited knowledge of the long-term effects associated with current treatment options [24]. In this study, all patients at Indrapura forefront hospital did not receive antiviral therapy, antibiotics, or corticosteroids. In this study, 6102 patients were included. Among them, 5923 patients (97.1%) recovered with appropriate management for those without symptoms and mild symptoms. Additionally, 36 patients (0.6%) chose to continue self-isolation outside the hospital. A total of 142 patients, accounting for 2.3% of the sample, were referred due to indications of desaturation, defined as a peripheral oxygen saturation (SpO2) level below 94%. One patient died due to a cardiovascular event. Furthermore, the analysis of clinical outcomes in relation to the length of stay (LOS) revealed that 74.5% of the total sample size ($n = 4529$) received treatment for less than 10 days, while 11.3% ($n = 692$) had an LOS of exactly 10 days, and 14.3% ($n = 871$) had an LOS exceeding 10 days. Furthermore, a total of 142 patients (2.3%) were referred due to indications of clinical deterioration, while 36 patients (0.6%) were in self-isolation.

According to the current guidelines from the World Health Organization (WHO), asymptomatic individuals with COVID-19 can end their isolation after 10 days from the initial PCR swab. For patients with symptoms, they can be released from isolation after 10 days from the onset of symptoms while being symptom-free for at least 3 consecutive days, without the need for another swab RT-PCR test [14]. Asymptomatic patients exhibit

a comparable viral load to symptomatic patients, suggesting that individuals without symptoms possess an equivalent capacity for transmitting the disease [27]. The primary obstacle in containing the transmission of COVID-19 among asymptomatic individuals is their lack of awareness regarding their potential to spread the virus to others [28]. The study reports that 42.2% of patients lacked direct contact with individuals diagnosed with COVID-19 [29]. In order to prevent and control the transmission of COVID-19, it is crucial to investigate the exposure history of asymptomatic patients, particularly among adolescents, women, and individuals without underlying health conditions who are more prone to being asymptomatic when infected. Additionally, suspected cases should undergo nucleic acid detection. Asymptomatic patients generally have a favorable prognosis. However, they should still be isolated and observed for potential transmission, as the duration of their disease is not shorter compared to symptomatic patients [30]. The significant number of asymptomatic individuals in our study reinforces the significance of social distancing, crowd avoidance, and patient isolation as preventive measures against disease transmission during the ongoing pandemic.

## 5. Conclusions

This study provides a comprehensive analysis of the clinical and epidemiological features observed in individuals infected with COVID-19 who exhibit either asymptomatic or mild to moderate symptoms. The available research evidence on asymptomatic infections is currently limited, and further investigation is required to elucidate the specific characteristics of these infections. Currently, isolation and close observation remain the preferred approaches for managing asymptomatic individuals with COVID-19.

**Author Contributions:** Conceptualization, E.A.T., J.W. and C.S.W.; methodology, D.A.P., N. and S.D.; software, A.A.H. and M.A.P.L.; validation, A.A.H. and M.A.P.L.; formal analysis, E.A.T., J.W. and C.S.W.; investigation, D.A.P., N., S.D., A.A.H., M.A.P.L., F.S., N.A.K.D., M.I.Z.A.R., T.H.W. and N.D.A.; resources, E.A.T., J.W. and C.S.W.; data curation, F.S., N.A.K.D., M.I.Z.A.R., T.H.W. and N.D.A.; writing—original draft preparation, D.A.P., N. and S.D.; writing—review and editing, A.A.H. and M.A.P.L.; visualization, A.A.H. and M.A.P.L.; supervision, E.A.T., J.W. and C.S.W.; project administration, F.S., N.A.K.D., M.I.Z.A.R., T.H.W. and N.D.A. All authors have read and agreed to the published version of the manuscript.

**Funding:** This research received no external funding.

**Institutional Review Board Statement:** The study was conducted in accordance with the Declaration of Helsinki, and approved by the Institutional Review Board (or Ethics Committee) of Universitas Airlangga School of Medicine (protocol code 37/EC/KEPK/FKUA/2021, approved 18 February 2021).

**Informed Consent Statement:** Not applicable.

**Data Availability Statement:** Data is unavailable due to privacy or ethical restrictions.

**Acknowledgments:** The authors wish to thank the chairman of the Indrapura forefront hospital, I.D.G. Nalendra Djaya Iswara and Djanuar Firtriadi for their support throughout the making of this paper.

**Conflicts of Interest:** The authors declare no conflict of interest.

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
