# Peer review of "Clinical Characteristics of 6102 Asymptomatic and Mild Cases for Patients with COVID-19 in Indonesia"

_pathophysiology, doi:10.3390/pathophysiology30030028_

Round 1

Reviewer 1 Report

The manuscript need gramatical revision and focusing on data re-analysis for douple cheking.

Need gramatical RV.

Author Response

Thank you for your comments. we have improved the English language used in the manuscript to the best of our ability.

Reviewer 2 Report

Generally a thorough and well described paper of value to epidemiologists and health system planners. Could health planning receive a bit more emphasis in the Discussion?

Abstract

line 18: making epidemiological studies....
line 24: Results
line 25: the total population was
line 39: spread of this disease

Results
line 178 received symptomatic therapy
lines 171 - 175 on re-swab data are not explored in the Discussion. How can the cure statistics be interpreted if so many (51.2%) tested positive after discharge?
Discussion
Font changes line 283 - line 311; line 237 - line 345
Why are the re-swab data not discussed?

A few minor comments on language highlighted above 

Author Response

Thank you for the comments. We have tried to improve the quality of the English language to the best of our ability, including the highlighted parts that you have mentioned.

Regarding your comment on: "Lines 171 - 175 on re-swab data are not explored in the Discussion. How can the cure statistics be interpreted if so many (51.2%) tested positive after discharge?"

The cure status was determined by the presence of clinical symptom reduction or absence after treatment, as well as two consecutive negative tests for the virus prior to discharge (Line 209-214). The re-swab test offers supplementary information of COVID-19 status subsequent to a negative result upon discharge.

The difference of font sizes has been resolved, however, we believe that this problem was due to the editing process and the problem  was not present in our original manuscript

Reviewer 3 Report

This paper presents a report on 6102 patients diagnosed as COVID-19 positive in Indonesia between 2020 and 2021. These patients were admitted to a single hospital facility, and the majority of them were asymptomatic or had mild symptoms of infection. The age group with the highest number of cases was people in their twenties. Hypertension was the most common underlying comorbidity. Due to the mildness of the symptoms, 74.5% of the patients remained hospitalized for less than 10 days. Patients with mild COVID-19 infection primarily experienced cough and hyposmia as symptoms. There were no specific drugs utilized to treat viral infections. The study's conclusion suggests that supportive care and isolation are crucial measures.

This study is an intriguing summary of a sizeable number of asymptomatic or mildly symptomatic COVID-19 patients in Indonesia - a scenario that has not been thoroughly examined before - which reports the current situation in the region. This study is unique as it summarizes the distinct characteristics of COVID-19 infection in Indonesia that differ from those of Europe, the U.S., or China, and thus will serve as a valuable reference for comparing findings in other countries. Accumulating similar studies and comparing them to future studies on pandemics caused by mutant virus strains could be valuable.

#1 Please clarify the reason why asymptomatic patients are being tested for COVID-19 infection.

#2 Please clarify the differences in age and length of hospital stay between asymptomatic patients and those with mild symptoms.

#3 Please indicate how many of the 191 patients with moderate symptoms were asymptomatic or mildly ill at the beginning of their hospitalization, and how many deteriorated after admission.

To consider this manuscript for publication, please add data on these three points and make minor revisions.

Reviewer 4 Report

The authors conducted a comprehensive investigation and the manuscript is generally well-addressed and well-written.

However, I have jusy one suggestions: form line 283 to 311, and from line 237 to 345: unifotm font size.

Author Response

Thank you for your comment. The difference between font sizes in the lines mentioned has been resolved.